# Study of the Spatial Spillover Effects of the Efficiency of Agricultural Product Circulation in Provinces along the Belt and Road under the Green Total Factor Productivity Framework

**Minghua Dai, Guanwei Wang \*, Jiaqiu Wang, Yuhan Gao and Quanzhen Lu**

School of Management, Dalian Polytechnic University, Dalian 116034, China; daiminghua2002@163.com (M.D.); jiaqiu01@outlook.com (J.W.); 18742022013@163.com (Y.G.)
\* Correspondence: 20072095137736@xy.dlpu.edu.cn

**Abstract:** In the context of China's socialist market economy, production and circulation are equally important. Production creates value, while circulation realises value, and both are essential components of socialised reproduction. This paper, based on panel data from 30 provinces and cities in China covering the period from 2010 to 2021, uses methods such as the slacks-based model (SBM), global Malmquist–Luenberger (GML) index, generalised method of moments with system estimation (system GMM) and spatial Durbin model to investigate the developing mechanism and influencing factors of agricultural product circulation efficiency in provinces along the Belt and Road Initiative (BRI) within the framework of green total factor productivity. This study found the following: First, the overall trend of green total factor productivity of agricultural product circulation in provinces along the BRI shows negative growth, especially after the launch of the BRI initiative in 2014. Second, the level of foreign investment has a positive impact on the green total factor productivity of agricultural product circulation in provinces along the BRI. On the other hand, environmental regulations, government support and industrial structure have negative impacts. Third, based on the spatial weight matrix of geographical adjacency, there is a positive spatial spillover effect on the green total factor productivity of agricultural product circulation.

**Keywords:** Belt and Road; green total factor productivity of agricultural product circulation; system GMM; spatial spillover





## 1. Introduction

As one of the world's largest producers and consumers of agricultural products, China is firmly committed to building a robust and efficient agricultural distribution system. This pursuit is not only a critical means to achieve the strategic goal of becoming an agricultural powerhouse but also an integral strategic component to completely realise agricultural modernisation in China. Particularly in the context of the Belt and Road Initiative launched in 2014, the concepts of green agriculture and sustainable agricultural development have gradually turned out to comprise one of the key criteria determining the international competitiveness of China's agricultural industry.

The Belt and Road Initiative aims to promote economic cooperation and connectivity across various regions including Asia, Europe and Africa. Since agriculture is the basic industry for most countries and regions along the BRI, this initiative places a strong emphasis on the agricultural sector, and agricultural products circulation plays a central role in facilitating regional economic integration and common progress. The provinces in China, as key stakeholders taking part in the "Belt and Road" Initiative, have different characteristics due to their heritage in traditional agricultural production, processing and marketing dynamics. Some regions have promising, good economic trajectories and advanced agrotechnological capabilities but finite resource endowments which prevent them from fully exploiting technological innovation as a competitive advantage. Conversely, other regions have abundant

agricultural resources but developmental disparities and inadequate transport and logistics infrastructure, which undermine the efficiency of the circulation of agricultural products. Within the expansive scope of the Belt and Road paradigm, there is a higher level of requirement for agricultural product circulation development among the provinces aligned with this initiative. In the existing agricultural product circulation framework, the intricate structural configurations, lengthy distribution channels, ambiguous distribution actors and suboptimal levels of organisational and informational integration collectively impede the rapid maturation of China's agricultural product circulation apparatus. This predicament poses challenges to seamless integration into the competitive global marketplace.

As the interplay of agrarian economic dynamics intensifies between provinces adjacent to China's Belt and Road trajectory and neighbouring nations and regions, the convergence of agrarian economic–cultural norms, agrotechnological infrastructure capabilities and agricultural development ideologies provides an opportune milieu for provinces to proactively recalibrate their prevailing agricultural product distribution system. The crux of this strategic recalibration lies in embedding green agricultural and sustainable agricultural development principles throughout the entire spectrum of agricultural product circulation. This strategic integration will improve the green TFP of agricultural product circulation itself and also promote cooperative development among interconnected provinces and territories traversing the Belt and Road trajectory.

Therefore, the present study aims to explore the field of agricultural product circulation efficiency development in provinces along China's Belt and Road route under the green TFP framework. Through a multifaceted analysis that includes environmental regulations, foreign direct investment, government support and other relevant dimensions, it is committed to discovering the correlation with the above factors. And then, this study seeks to elucidate whether these regions show signs of spatial spillovers, anchored in the framework of promoting the development of green TFP of agricultural products circulation in provinces along China's Belt and Road corridor.

## 2. Literature Review and Theoretical Analysis

### 2.1. Literature Review

In terms of researching the green development of agricultural product circulation, the agricultural product circulation industry represents a leading sector in the design of the agricultural industry and occupies an equally important position in agricultural production. Improving both the circulation efficiency and production efficiency is an essential means of promoting the high-quality development of the agricultural industry. The aim of agricultural product circulation is to efficiently transform producers' agricultural products into products needed by consumers and to ensure the satisfaction of consumers' demands. In the existing agricultural product circulation system, the optimisation of the circulation system and the adjustment of the circulation structure and mode are the key issues of domestic and foreign scholars. However, with the introduction of green agriculture and the concept of sustainable agricultural development, the questions of how to incorporate the concept of green development into the agricultural product circulation system and how to promote the application of green innovative technologies in the field of agricultural product circulation still need further exploration and solutions [1–3]. As a bridge between agricultural production and consumption, the circulation of agricultural products advocates green development, which can better optimise the circulation channels of agricultural products and further promote the overall development of green agricultural production efficiency and product quality [4,5]. In view of unprecedented global transformation, Wang [6] suggests that China has entered a stage of high-quality development, shifting from "high-carbon growth" to "green growth", and emphasises strengthening the construction of the circulation system in the circulation link, promoting a higher level of openness to the outside world. Zhao [7] et al. establish an intermediary effect model in the labour market to explore the growth path of China's agricultural product circulation industry. Qi [8] et al. suggest speeding up the construction of agricultural product quality

standards, promoting the scale development of agricultural product circulation channels and, thus, improving the efficiency of the supply chain.

In the study of green total factor productivity (GTFP) in the circulation of agricultural products, GTFP is a modification and measurement of traditional total factor productivity that takes into account environmental factors. It aims to address the problem that the traditional indicators of total factor productivity neglect environmental externalities. GTFP takes into account the efficiency of resource use and brings environmental sustainability into the scope of productivity considerations, making it crucial for assessing whether economic growth is sustainable. In measuring GTFP, scholars at home and abroad have, in recent years, widely used data envelopment analysis (DEA) to quantify the extent of resource and environmental use in production activities. Tone [9] introduced a nonradial, nonangular distance function, which allows a better measurement of green total factor productivity. The circulation sector of agricultural products includes the upstream, midstream and downstream sectors of the agricultural industry. By measuring green total factor productivity in agricultural product circulation, we can more comprehensively reflect the existing shortcomings in the development of China's agricultural product circulation system. In previous research on evaluating the efficiency of the agricultural product cycle, many scholars at home and abroad did not consider the environmental factors and pollution caused by the agricultural product cycle industry. Instead, they focused more on studying efficiency from the perspective of expected outputs and inputs. Zhang [10] et al. constructed an efficiency measurement system for agricultural product circulation that includes four aspects—circulation scale, efficiency, speed and cost—and 11 basic indicators. They divided the development of China's agricultural product circulation efficiency into four main stages and emphasised that the integration of agricultural products and e-commerce will be an important direction for future development. Focusing on the western region of China, Cheng [11] et al. used the Malmquist index analysis method to measure the efficiency of agricultural product circulation and proposed relevant improvement suggestions. Wang [12] et al. used DEA to formulate the main ideas and framework for evaluating the efficiency of agricultural product circulation and conducted empirical analysis. With the rapid development of China's rural economy and the acceleration of urbanisation, especially in the context of the green economy and low-carbon industrial transformation, it is essential to incorporate a green circulation evaluation system into the construction of agricultural product circulation efficiency indicators. This approach should comprehensively consider the actual impact of environmental factors in the agricultural product circulation process and seek a development path for a green agricultural product circulation system that is suitable for China's market economy conditions [13,14].

In the study of the factors influencing the efficiency of the circulation of agricultural products within the framework of the green total factor productivity (GTFP), there are two main aspects to be taken into account. On the one hand, the inherent characteristics of agricultural products, such as seasonality, timeliness and vulnerability, often prevent them from fully participating in market competition. On the other hand, the low level of informatisation and organisation of the entities involved in the circulation of agricultural products, represented in particular by farmers, contributes to the inefficiency of the circulation of agricultural products. In previous studies, scholars have generally focused on the impact of factors such as agricultural product prices, government financial support and agricultural product transport infrastructure on the efficiency of agricultural product circulation. Although this has improved the development of agricultural product circulation efficiency to some extent, it has also led to resource waste and duplication. Li [15] et al. analysed the main influencing factors on the efficiency of agricultural product circulation in the Beijing–Tianjin–Hebei region from an input–output perspective. They found that factors such as technological innovation capacity, development of circulation units, construction of circulation carriers and optimisation of the circulation environment have impacts on the efficiency of agricultural product circulation in this region. With the progress of green development in the circulation of agricultural products, the need for market regulation

and policy support from the government and relevant departments becomes more obvious. Therefore, the selection of appropriate environmental regulation measures to promote the improvement of agricultural green total factor productivity plays an important role in the transformation and upgrading of agricultural modernisation. Huang [16] et al. used the Malmquist–Luenberger index from the perspective of environmental regulation to measure the green total factor productivity of wheat. They found that China's wheat industry has experienced extensive growth at the expense of the ecological environment. Hu [17] et al. measured provincial-level panel data in China from 2007 to 2018 and found that the improvement of environmental regulations can promote the development of agricultural total factor productivity, and there is a significant spatial spillover effect between the two. Zhan [18] et al. explored the causal relationship between agricultural green productivity and food security from the perspective of different environmental regulations. They found that appropriate environmental policies can promote the growth of agricultural green productivity and contribute to the sustainable development of food security.

### 2.2. Research Hypotheses

The environmental Kuznets curve [19] shows that in the early stages of growing GDP, there is environmental degradation. However, as the economy develops to a certain extent, environmental damage gradually decreases and environmental quality improves. This curve implies that in the early stages of economic growth, resource exploitation and pollution often increase. However, as the economy develops and technology advances, people gradually become aware of the importance of environmental protection and put more of their efforts into green technological innovation, thus achieving a win-win situation for economic growth and environmental protection. In the field of agricultural product circulation, by adopting green technologies, agricultural product circulation enterprises can increase the scale of production and adopt more efficient production and distribution methods. For example, the use of intelligent logistics systems and energy-saving transportation vehicles can not only reduce costs but also improve transportation efficiency. At the same time, the use of environmentally friendly packaging materials can reduce resource waste and lower packaging costs. These measures can help agribusinesses reduce production and operating costs, improve efficiency and achieve economies of scale [20,21].

From the perspective of environmental economics, moderate environmental regulations can provide incentives for companies in the agricultural product chain to adopt environmentally friendly measures, reduce resource waste, optimise production processes and improve environmental efficiency. Although environmental regulations may increase costs for companies, companies can achieve a win-win situation of environmental and economic benefits through technological innovation and resource optimisation. However, based on Michael Porter's competitive hypothesis proposed in 1995 [22], environmental regulations play a dual role in influencing the green total factor productivity in the circulation of agricultural products. On the one hand, strict environmental regulations may force enterprises involved in the circulation of agricultural products to raise their environmental standards, thereby increasing their operating costs, which could temporarily restrain the development of total factor productivity in the circulation of agricultural products. On the other hand, environmental regulations may also drive agricultural product circulation enterprises to transform and upgrade by adopting more environmentally friendly and efficient technologies and management measures, optimising resource utilisation, reducing costs and promoting the long-term improvement of green total factor productivity in agricultural product circulation.

Based on this, this paper proposes Hypothesis 1:

**Hypothesis 1 (H1).** *In China's existing agricultural product circulation system, the intensity of environmental regulations will reduce the development of green total factor productivity in agricultural product circulation.*

The proposal of the Belt and Road Initiative has had a positive impact on the development of the agricultural industry in China's provinces and regions along the Belt and Road, especially in the area of agricultural product circulation, bringing both greater opportunities and challenges. On the one hand, due to the uneven level of economic development among these provinces, there are differences in agricultural production levels, transportation infrastructure and government support. On the other hand, with the gradual increase in foreign investment and the continuous opening up of the agricultural market, some economically backward regions may prioritise increasing agricultural production efficiency at the expense of the ecological environment, neglecting the aspect of agricultural sustainability. Li [23] et al. focused on China's export trade, believing that the Belt and Road Initiative will help boost export trade in regions along the Belt and Road. By using the double-difference method to evaluate the policy, they found that its promotion effect does not have a stable and sustainable dynamic effect and varies from region to region. Zhang [24] et al. also showed that foreign trade can have an impact on the green total factor productivity of provinces along the Belt and Road. On the other hand, as important participants in the initiative, China's provinces and regions along the Belt and Road can introduce advanced agricultural production technologies, agricultural product quality control techniques and agricultural product circulation management experiences from other countries and regions. At the same time, they can gain broader opportunities for cooperation in agricultural product markets, thereby enhancing the competitiveness of local agricultural products and promoting the development of agricultural product circulation in surrounding cities and regions.

Based on this, the present study proposes Hypothesis 2:

**Hypothesis 2 (H2).** *Under the framework of green total factor productivity, there is a "spatial spillover effect" in the development of agricultural product circulation efficiency in China's provinces and regions along the Belt and Road.*

## 3. Research Methods and Data Sources

### 3.1. SBM-DEA Model

Green total factor productivity refers to the ability to maximise production efficiency, taking into account environmental, resource and economic factors while utilising existing resources. Data envelopment analysis (DEA) is a commonly used method for measuring efficiency with multiple inputs and outputs. However, DEA models have certain limitations because they do not take into account potential resource wastage and output shortfalls, which can lead to biased assessment results. To address this issue, scholars such as Chung [25] et al. proposed the Malmquist–Luenberger index analysis method based on the directional distance function (DDF). Although this method effectively addresses missing values for undesirable outcomes in traditional DEA measurement models, DDF models based on radial or angular distances still have shortcomings. First, the radial distance function can only ensure that the ratio of changes between desirable and undesirable outcomes is equal. In the presence of nonzero slack, the results of the radial-distance-based method will overestimate production efficiency. Secondly, the angle-based distance function requires the selection of a measurement angle, and calculations based on either output or input angles will produce biased results by neglecting the other angle.

To overcome the limitations of DDF models, this study uses the slacks-based model (SBM), which is based on nonradial, nonangular directional distance functions. The SBM model provides a more comprehensive and accurate measure of green production efficiency by simultaneously considering multiple input and output factors and effectively modelling slack factors. By introducing slack variables, the SBM model quantifies potential resource waste and output shortfalls in decision units, leading to a more accurate assessment of green total factor productivity.

Suppose there are $K$ ($K = 1, 2, , \ldots, k$) decision-making units (DMUs) in the sample under consideration. Each DMU has $N$ types of inputs $X = \{x_1, x_2, \ldots, x_n\} \in R_+^N$ in the

agricultural product circulation process and can produce $M$ types of desirable outputs $Y = \{y_1, y_2, \ldots, y_n\} \in R_+^M$ and $Q$ types of undesirable outputs $Z = \{z_1, z_2, \ldots, z_n\} \in R_+^Q$. Following the research approach of Tone [9] et al., a nonradial, nonangular slack-based model (SBM) directional function can be constructed that includes both desirable outputs $M$ and undesirable outputs $Q$:

$$\rho^* = min \frac{1 - \left[\frac{1}{N}\sum_{n=1}^{N}\frac{s_n^x}{x_n^k}\right]}{1 + \frac{1}{M+Q}\left[\sum_{m=1}^{M}\frac{s_m^y}{y_m^k} + \sum_{q=1}^{Q}\frac{s_q^z}{z_q^k}\right]} \tag{1}$$

$$\text{s.t} \begin{cases} X_n^k = \sum_{k=1}^{K}\lambda_k x_n^k + s_n^x \\ Y_m^k = \sum_{k=1}^{K}\lambda_k y_m^k + s_m^y \\ Z_q^k = \sum_{k=1}^{K}\lambda_k z_q^k + s_q^z \\ s_n^x \geq 0, s_m^y \geq 0, s_q^z \geq 0, \lambda_k \geq 0 \end{cases}$$

In Equation (1), $s^x$, $s^y$, $s^z$ represent input slack variables (measuring input excess), desirable output slack variables (measuring output deficiency) and undesirable output slack variables (measuring undesirable output deficiency), respectively. $\lambda_k$ represents the average distance of each DMU's actual inputs and outputs to the production frontier, indicating the level of input and output inefficiency. The objective function $\rho^* \in [0,1]$, where $\rho^* = 1$ indicates that the DMU is operating at full efficiency, while $\rho^* < 1$ indicates the presence of efficiency losses, suggesting that improvements in both inputs and outputs are possible.

### 3.2. Analysis of the GML Index

The global Malmquist–Luenberger (GML) index is an extended form of the Malmquist–Luenberger (ML) index. The calculation of total factor productivity requires the use of index methods. Traditional ML indices lack transitivity. Therefore, this study chooses the GML index to measure the temporal variation and technological progress of green total factor productivity in agricultural product circulation (GTFPCH). Based on the findings of Oh's [26] research, the GML index can be formulated as follows:

$$GML^{t,t+1}\left(x^{t+i}, y^{t+i}, z^{t+i}; x^t, y^t, z^t\right) = \frac{1 + D^G\left(x^t, y^t, z^t\right)}{1 + D^G\left(x^{t+i}, y^{t+i}, z^{t+i}\right)} \tag{2}$$

When the GML index is greater than 1, it indicates an increase in expected outputs and a decrease in non-expected outputs, leading to an improvement in green total factor productivity. Conversely, when the GML index is less than 1, it indicates an increase in non-expected outputs and a decrease in expected outputs, resulting in a decline in green total factor productivity.

The GML index can be further decomposed as follows (Appendix A): GTFPCH = Global Technical Efficiency Change (GTC) × Global Efficiency Change (GEC), where GEC is the degree of proximity of actual production points to the production frontier and GTC is the degree of outward expansion of the production frontier.

### 3.3. Dynamic-Panel-Model-Based Empirical Analysis

In the actual development process, the development level of GTFP in the circulation of agricultural products has lagged characteristics. The development level of GTFP in the current period may be influenced by the development level of GTFP in the previous period. Therefore, the lagged GTFP development level $GTFPCH_{i(t-1)}$ is introduced into the model as an explanatory variable to establish a dynamic panel model.

$$GTFPCH_{it} = \alpha_i + \beta GTFPCH_{i(t-1)} + \theta X_{it} + u_i + \varepsilon_{it} \tag{3}$$

In this context, $i$ represents the region, $t$ represents time and $GTFPCH_{it}$ represents the level of development of green total factor productivity (GTFP) in the circulation of

agricultural products in region $i$ in year $t$. $X_{it}$ represents a set of explanatory variables that may affect the level of development of GTFP in the circulation of agricultural products. $\alpha_i$ denotes the intercept, $u_i$ represents the unobservable individual fixed effects of region i and $\varepsilon_{it}$ is the error term.

When running a dynamic panel model regression, a challenge arises from the inclusion of the lagged dependent variable in Equation (3). When estimating the results using traditional ordinary least squares (OLS) regression or fixed effects models, the endogeneity problem associated with the lagged dependent variable within the panel effects can lead to biased estimation results. Therefore, this study uses the generalised method of moments (GMM) approach to perform dynamic panel model regression analysis. The aim is to address the endogeneity problem inherent in the variables and thus obtain more effective estimation results. The application of the system GMM method provides a robust framework to address potential endogeneity concerns, ensuring that parameter estimation remains unbiased and efficient. By mitigating the endogeneity problem in the dynamic panel model, this methodology enhances the reliability and accuracy of the regression analysis, facilitating a more rigorous investigation of the intricate dynamics of green total factor productivity within the realm of agricultural product circulation.

### 3.4. Global Moran's I Index

The global Moran's I index is a commonly used spatial autocorrelation analysis method used to measure the spatial correlation of variables. Using the global Moran's I index, we can determine whether the green total factor productivity of the agricultural product cycle and each explanatory variable have spatial clustering or dispersion tendencies, and we can reveal the spatial dependency relationships among provinces. Where n represents the number of spatial units, $X_i$ and $X_j$ are the observed values of green total factor productivity (GTFP) in agricultural product circulation for provinces $i$ and $j$, respectively. $W_{ij}^{\rho}$ represents the spatial weight matrix, which reflects the spatial agglomeration level of GTFP in agricultural product circulation among provinces along the Belt and Road Initiative (BRI) through spatial correlation analysis. In this regard, following Wu [27], this study constructs the spatial weight matrix of geographical adjacency.

$$
\begin{gathered}
Global\,Moran's\ I = \frac{\sum_{i=1}^{n}\sum_{j=1}^{n}W_{ij}^{\rho}\left(X_t-\overline{X}\right)\left(X_j-\overline{X}\right)}{S^2\sum_{i=1}^{n}\sum_{j=1}^{n}W_{ij}^{\rho}} \\
S^2 = \frac{1}{n}\sum_{i=1}^{n}\left(X_i-\overline{X}\right)^2 \\
\overline{X} = \frac{1}{n}\sum_{i=1}^{n}X_i
\end{gathered}
\tag{4}
$$

### 3.5. Modelling Spatial Measurements

The green total factor productivity (GTFP) of agricultural products circulating within the same region is influenced not only by factors such as the intensity of environmental regulations and the level of foreign direct investment but also by spatial effects on closely connected regions. Therefore, following the research of Lesage [28] et al., this study constructs a spatial Durbin model (SDM).

$$
GTFPCH_{it} = \theta X_{it} + \beta_1 W \times GTFPCH_{it} + \beta_2 W \times \theta X_{it} + u_i + \varepsilon_{it}
\tag{5}
$$

In the SDM, $\beta_1$ and $\beta_2$ are the spatial autoregressive coefficients and W is the spatial weight matrix. The definitions of the other variables are the same as in Formula (3).

### 3.6. Variable Selection and Sources

3.6.1. Explained Variable

The explained variable in this study is the green total factor productivity in agricultural product circulation (GTFPCH). An evaluation index system is constructed based on the SBM model and GML index analysis. The selection of input and output variables is guided

by factors including capital input, labour and transport infrastructure. The selected input variables are capital input, labour force and transport infrastructure. For the expected output variables, the total value of agricultural products in circulation and the volume of agricultural products imported and exported are selected. Unexpected output variables include the total amount of carbon dioxide emissions and chemical oxygen demand (COD) generated during the agricultural product cycle. The specific selection of indicators is outlined below (Table 1).

**Table 1.** Assessment indicator system for green total factor productivity of agricultural product cycle based on input–output perspective.

| Type of Indicator | Content of Indicator | Description of Indicator | Symbol | Unit |
|---|---|---|---|---|
| Inputs | Circulation of agricultural products fixed capital | Fixed capital input of transport, storage and postal services, wholesale and retail trade, hotels and restaurants × Final consumption rate × Share of household consumption in final consumption × Engel coefficient (national average) | $x_1$ | CNY 100 million |
| | Number of enterprises involved in the circulation of agricultural products | Number of enterprises involved in the circulation of agricultural products from the Wind and Tonghuashun databases | $x_2$ | Units |
| | Development of transport infrastructure | Total kilometres of railways, motorways and waterways | $x_3$ | Kilometre (km) |
| Desired outputs | Circulation of agricultural products gross output value | Gross output value of transport, storage and postal services, wholesale and retail trade, accommodation and food services × Final consumption rate × Share of household final consumption in final consumption × Engel coefficient (national average) | $y_1$ | CNY 100 million |
| | Total import and export trade volume of agricultural products | Total import and export trade volume of agricultural products in different regions from statistical yearbooks | $y_2$ | USD 10,000 |
| Undesired outputs | $CO_2$ and COD emissions in the circulation stage of agricultural products | Total $CO_2$ and COD emissions from transport, storage and postal services, wholesale and retail trade, accommodation and food services | $fy_1$ | Ten thousand metric tonnes (MT) |

3.6.2. Explanatory Variable

This study selects six indicators to observe the influence of various factors on GTF-PCH in the Belt and Road Initiative provinces. The selected indicators are the lagged development level of GTFPCH, industrial structure, environmental regulation, government support, foreign direct investment (FDI) and agricultural product prices. When examining the correlation between the current GTFPCH and its lagged period, to avoid the problem of overidentification in model estimation, the choice of the lagged period should be less than one-third of the total study period, following the study of Holtzeakin [29] et al. Therefore, this paper systematically sets the GTFPCH lagged by one period as a variable to explore its impact on the development of agricultural product flow green total factor productivity. Here, GTFPCH is the dependent variable, and the explanatory variables include the lagged development level of GTFPCH (L.GTFPCH), environmental regulation (ER), government support (GP), foreign direct investment (FDI), industrial structure (ST) and agricultural product prices (AP). The specific settings of these indicators are as follows (Table 2).

**Table 2.** Explanation of variable selection.

| Type of Indicator | Content of Indicator | Description of Indicator | Symbol | Unit |
|---|---|---|---|---|
| | The lagged development level of green total factor productivity of China's agriculture | Calculated results of the study | L.GTFPCH | / |
| | Government support | Government fiscal expenditure in the relevant aspects of agricultural product circulation | GP | CNY 100 million |
| Explanatory variable | Foreign investment level | Foreign direct investment (FDI) as a percentage of GDP | FDI | CNY |
| | Environmental regulation | The number of environmental protection proposals in the two sessions | ER | Units |
| | Industrial structure | Total value of primary industry/Total value of secondary and tertiary industry | ST | CNY |
| | Agricultural product prices | Price index for food in retail trade (previous year = 100) | AP | / |

### 3.6.3. Variable Descriptive Statistics

This section of the paper provides brief descriptive statistics for all explanatory variables, explained variables, input variables and output variables (Table 3).

**Table 3.** Descriptive statistics of the variables.

| Type of Indicator | Symbol | Content of Indicator | Sample Size | Mean | Standard Deviation | Min. | Max. |
|---|---|---|---|---|---|---|---|
| | $x_1$ | Circulation of agricultural products fixed capital | 360 | 236.166 | 154.939 | 14.740 | 700.782 |
| Inputs | $x_2$ | Number of enterprises involved in the circulation of agricultural products | 360 | 1072.944 | 1045.098 | 46 | 6698 |
| | $x_3$ | Development of transport infrastructure | 360 | 159,915.1 | 84,839.11 | 14,584.39 | 467,973.3 |
| | $y_1$ | Circulation of agricultural products gross output value | 360 | 456.613 | 374.784 | 18.125 | 1776.084 |
| Desired outputs | $y_2$ | Total import and export trade volume of agricultural products | 360 | 401.467 | 654.043 | 0.533 | 3795.685 |
| Undesired outputs | $fy_1$ | $CO_2$ and COD emissions in the circulation stage of agricultural products | 360 | 428.465 | 586.451 | 4.963 | 3106.204 |
| Explained variable | GTFPCG | Green total factor productivity of agricultural product circulation | 204 | 0.996 | 0.087 | 0.613 | 1.560 |
| | L.GTFPCH | The lagged development level of green total factor productivity of China's agriculture | 187 | 0.997 | 0.090 | 0.613 | 1.560 |
| Explanatory variable | ER | Environmental regulation | 204 | 405.270 | 365.176 | 16.000 | 2471.000 |
| | GP | Government support | 204 | 827.693 | 439.360 | 123.368 | 3009.98 |
| | FDI | Foreign investment level | 204 | 56.437 | 73.883 | 0.045 | 290.400 |
| | ST | Industrial structure | 204 | 0.890 | 0.059 | 0.742 | 0.997 |
| | AP | Agricultural product prices | 204 | 105.758 | 4.156 | 97.000 | 122.500 |

Source of data: actual measurements.

### 3.6.4. Data Sources

This study uses a sample of 30 provinces and cities in China for the period 2010 to 2021. The sample data are obtained from the Statistical Yearbook of China and corresponding provincial and municipal yearbooks, the China Environmental Statistical Yearbook, carbon

accounting databases, the Wind database, the Tonghuashun database and the China Rural Statistical Yearbook. Linear interpolation was used to complete the data where gaps occurred during the data collection process. The Tibet region was excluded from the empirical analysis in this study due to incomplete data and lack of public data.

## 4. Development Characteristics of Green Total Factor Productivity in the Circulation of Agricultural Products in Provinces along the Belt and Road

Based on the aforementioned input–output indicators, this study employs Stata 16.0 software to measure the green total factor productivity of agricultural product circulation in 30 provinces and municipalities (excluding Tibet) in China. The geometric method of the annual green total factor productivity index of agricultural product circulation in the 30 provinces and municipalities is taken as the national green total factor productivity index of agricultural product circulation for that year. The geometric mean of the annual green total factor productivity index of agricultural product circulation in the Belt and Road Initiative (BRI) provinces is taken as the green total factor productivity index of agricultural product circulation for the BRI provinces for that year.

### 4.1. Green Total Factor Productivity in the Agricultural Product Circulation of Provinces along the Belt and Road Initiative from a National Perspective

Overall, from 2010 to 2021, the green total factor productivity in the agricultural product circulation of provinces along the Belt and Road Initiative, viewed from a national perspective, exhibited a negative growth trend. Moreover, the average annual index of green total factor productivity in the agricultural product circulation of the Belt and Road provinces was lower than the national average. Specifically, the average annual index of green total factor productivity in the agricultural product circulation of the entire country from 2010 to 2021 was 0.9915, with an average annual decline rate of 0.85%. For the Belt and Road provinces during the same period, the average annual index of green total factor productivity in the agricultural product circulation was 0.9923, with an average annual decline rate of 0.77% (Figure 1).

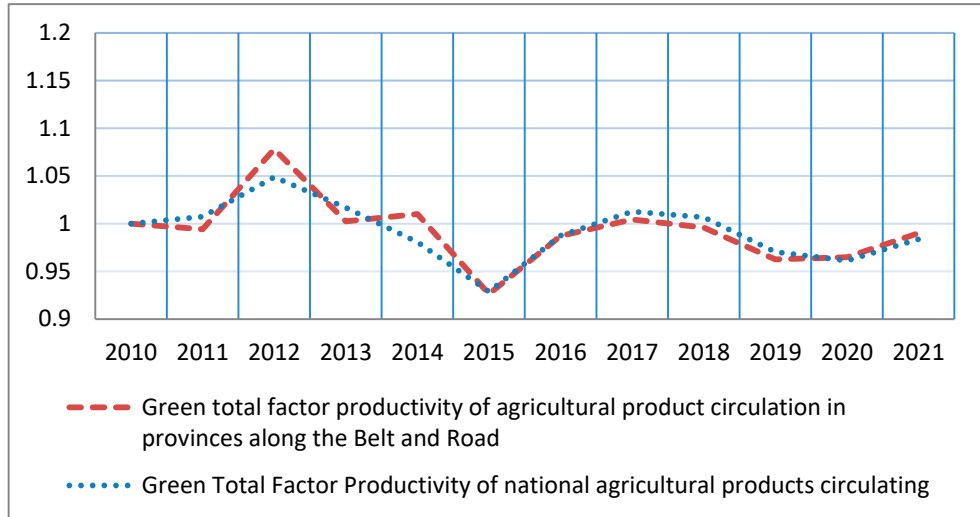

**Figure 1.** Green total factor productivity in the agricultural product circulation of provinces along the Belt and Road Initiative from a national perspective. Source of data: actual measurements.

From the changes in the index of green total factor productivity in agricultural product circulation, the overall situation is not optimistic. Moreover, from 2010 to 2021, the index of green total factor productivity in agricultural product circulation of provinces along the Belt and Road Initiative was mostly lower than the national index, and it exhibited obvious fluctuations. Specifically, in the years 2011, 2015, 2016, 2018, 2019, 2020 and 2021, the index of green total factor productivity in agricultural product circulation of the Belt and Road

provinces was below 1, indicating a deterioration in the green total factor productivity of agricultural product circulation. In the remaining years, the green total factor productivity in agricultural product circulation showed some improvement to a certain extent. After 2019, the provinces along the Belt and Road Initiative experienced an upward trend in green total factor productivity in agricultural product circulation.

### 4.2. Green Total Factor Productivity in Agricultural Product Circulation in Provinces Along the Belt and Road Initiative

For provinces along the Belt and Road Initiative, China acts as both the promoter and participant in the initiative. As one of its important economic pillars, agriculture has an apparent impact on the development of green total factor productivity in agricultural product circulation. The initiative was proposed at the end of 2013, and this study analyses the development of green total factor productivity in agricultural product circulation in provinces along the Belt and Road Initiative during two periods: preinitiative (2010–2013) and postinitiative (2014–2021) (Table 4).

**Table 4.** Green total factor productivity in agricultural product circulation in provinces along the Belt and Road Initiative.

| Province | 2010–2013 | | | 2014–2021 | | |
|---|---|---|---|---|---|---|
| | **GTFPCH** | **GTC** | **GEC** | **GTFPCH** | **GTC** | **GEC** |
| Inner Mongolia | 1.098 | 1.110 | 0.988 | 0.916 | 1.000 | 0.916 |
| Liaoning | 1.054 | 1.069 | 0.986 | 0.960 | 0.957 | 1.003 |
| Jilin | 0.972 | 0.964 | 1.008 | 1.000 | 1.023 | 0.978 |
| Heilongjiang | 1.072 | 1.107 | 0.968 | 0.929 | 1.000 | 0.929 |
| Shanghai | 1.002 | 1.000 | 1.002 | 1.000 | 1.000 | 1.000 |
| Zhejiang | 1.035 | 0.968 | 1.069 | 0.981 | 0.975 | 1.006 |
| Fujian | 1.015 | 1.005 | 1.009 | 0.984 | 0.984 | 1.000 |
| Guangdong | 1.029 | 0.996 | 1.033 | 0.954 | 0.970 | 0.983 |
| Guangxi | 1.002 | 1.013 | 0.988 | 0.952 | 0.954 | 0.999 |
| Hainan | 1.047 | 0.971 | 1.078 | 0.994 | 1.036 | 0.959 |
| Chongqing | 1.031 | 1.022 | 1.009 | 0.985 | 0.995 | 0.990 |
| Yunnan | 1.028 | 1.047 | 0.982 | 0.970 | 0.960 | 1.010 |
| Shaanxi | 1.003 | 0.986 | 1.017 | 0.994 | 1.002 | 0.992 |
| Gansu | 1.013 | 1.027 | 0.986 | 0.979 | 0.978 | 1.001 |
| Qinghai | 1.008 | 1.004 | 1.003 | 1.001 | 1.024 | 0.978 |
| Ningxia | 0.952 | 0.947 | 1.005 | 0.988 | 0.995 | 0.993 |
| Xinjiang | 1.002 | 0.999 | 1.004 | 1.001 | 1.004 | 0.997 |
| Mean | 1.021 | 1.013 | 1.008 | 0.976 | 0.991 | 0.984 |

Source of data: actual measurements.

Overall, before the proposal of the Belt and Road Initiative (2010–2013), the green total factor productivity index in agricultural product circulation in the provinces along the Belt and Road was 1.021, with an average annual growth rate of 2.1% over the four-year period. After the proposal of the Belt and Road Initiative (2014–2021), the green total factor productivity index in agricultural product circulation in these provinces was 0.976, with an average annual decline of 2.4% over the seven-year period.

Looking at individual provinces and cities during the preproposal period (2010–2013), out of the 17 provinces and cities along the Belt and Road, 15 provinces and cities achieved effective growth in green total factor productivity in agricultural product circulation. Only Jilin and Ningxia experienced a decline in green total factor productivity, showing a downward trend. After the proposal of the Belt and Road Initiative (2014–2021), only four provinces and cities, namely, Jilin, Shanghai, Qinghai and Xinjiang, achieved effective growth in green total factor productivity in agricultural product circulation. Among them, only Xinjiang and Qinghai achieved effective growth in green total factor output change, while the remaining 13 provinces and cities experienced a deterioration in green total factor productivity change, showing a downward trend.

From the perspective of efficiency composition, the green total factor productivity in agricultural product circulation (GTFPCH) can be expressed as the product of green technology change index (GTC) and green technology efficiency index (GEC). After the proposal of the Belt and Road Initiative in 2014, the GTFPCH in the provinces along the Belt and Road decreased by 4.5%, with GTC declining by 2.2% and GEC declining by 2.4%. The mean values of both GTC and GEC were less than 1, indicating a deterioration in efficiency.

At this critical moment of transformation in modern agricultural development, the level of green technology plays a crucial role in establishing a sound modern agricultural product circulation system and ensuring the high-quality development of agricultural product circulation. Technological backwardness can lead to issues, for instance, redundant capital, redundant transportation infrastructure and redundant labour in the circulation process. It can also result in wastage of resources, inefficiency and low quality in the production process, further reducing the development of green total factor productivity in agricultural product circulation. Additionally, Appendix B shows that the GTFP in Inner Mongolia, Jilin, Heilongjiang, Hainan and Qinghai provinces is mainly affected by low green technology efficiency, especially in Inner Mongolia and Heilongjiang, where the agricultural green technology progress index is significantly higher than the technology application efficiency. This indicates that these two provinces have not effectively translated the achievements of agricultural green technology progress into the production and circulation field.

To improve the development of green total factor productivity in agricultural product circulation, it is important to promote green technology progress and enhance the efficiency of agricultural technology application. This can be achieved by improving agricultural product circulation efficiency, optimising the circulation system, enhancing circulation quality and services and strengthening the competitiveness of agricultural products. It is necessary to explore and innovate in technology research and development, resource guarantee, environmental governance and green technology application to enhance the level of green total factor productivity in agricultural product circulation in the provinces along the Belt and Road Initiative.

## 5. An Empirical Analysis of the Green Total Factor Productivity of the Circulation of Agricultural Products in the Provinces along the Belt and Road

*5.1. Dynamic-Panel-Model-Based Empirical Analysis*

In this study, the dynamic generalised method of moments (system GMM) is used as the estimation method for the analysis of the model. The regression results are presented as follows (Table 5).

**Table 5.** Results of the system GMM regression.

| Variable | System GMM Model | |
|---|---|---|
| | Coefficient (Z-Value) | *p*-Value |
| L.GTFPCH | −0.1139 *** (−1.840) | 0.003 |
| ER | −0.00001 *** (−2.79) | 0.005 |
| GP | −0.00003 *** (−6.75) | 0.000 |
| FDI | 0.0002 *** (4.72) | 0.004 |
| ST | −0.1087 ** (−2.48) | 0.013 |
| AP | −0.0011 ** (−2.06) | 0.039 |
| Constant | 1.3436 *** (11.23) | 0.000 |
| Wald statistic | 6,800,000 | |
| Wald associated probability | 0.000 | |
| Arellano–Bond (1) | 0.091 | |
| Arellano–Bond (2) | 0.174 | |
| Sargan tests | 0.268 | |
| Hansen tests | 1.000 | |

Note: **, *** denote significant at 5% and 1% levels.

First, for the GTFPCH equation, the Wald test statistic and associated probability are important at the 1% level, rejecting the null hypothesis that all explanatory variables are equal to zero. Second, in the Arellano–Bond autocorrelation test, the *p*-value for AR (1) is 0.091 and for AR (2) is 0.174, indicating acceptance of the null hypothesis of "no serial

correlation in the disturbances", suggesting the absence of second-order serial correlation in the error terms. The *p*-values for the Sargan and Hansen tests are 0.268 and 1.000, respectively, indicating acceptance of the null hypothesis that all instruments are valid, confirming the absence of an overidentification problem with the instruments. These results indicate that the system GMM model is appropriately specified and the estimation results are highly reliable.

L.GTFPCH is negatively correlated with the green total factor productivity of agricultural products, and it passes the significance test at the 1% level. On the one hand, the economic development levels of the provinces along the Belt and Road are different, and some provinces may sacrifice the ecological environment to accelerate the improvement of agricultural product circulation efficiency, thereby reducing the development of agricultural product green circulation efficiency. On the other hand, due to the time sensitivity of agricultural product circulation, the implementation of environmental regulations and government support may be delayed in some regions, leading to a delay in the development of green total factor productivity in agricultural product circulation. Finally, the inadequate and complex structure of the agricultural product circulation system, as well as the redundancy and duplication of circulation functions in the provinces along the Belt and Road, are fundamental obstacles to the development of green circulation.

There is a negative correlation between environmental regulations and green total factor productivity of agricultural products, and it passes the significance test at the 1% level, verifying Hypothesis 1. On the one hand, tightening environmental regulations will increase the operating costs of agricultural product circulation enterprises. As enterprises engage in the production, circulation and sale of agricultural products, they need more technology, equipment and manpower to meet environmental requirements. In this case, the development space of these enterprises will be limited, which will have a certain negative impact on their green total factor productivity. On the other hand, this study uses the number of environmental protection proposals during the two sessions as a measure of environmental regulation. There may be a time lag between the formulation and implementation of environmental policies. Spatially, the development levels of different provinces and regions are uneven, which requires tailor-made approaches. Therefore, environmental regulatory measures should be based on the actual development of agricultural product circulation in the provinces along the Belt and Road, and specific analysis should be conducted for each problem. While fully safeguarding the interests of agricultural product circulation entities, which are mainly represented by farmers, efforts should be made to cultivate talent who understand and love agriculture and involve them in the fields and farmlands. By integrating the concepts of green development and sustainability into the entire agricultural product circulation process from upstream to downstream, we can improve the development of green total factor productivity of agricultural products.

Government support is negatively correlated with green total factor productivity in the circulation of agricultural products, a relationship confirmed by a significant test at the 1% level. On the one hand, in the context of traditional agricultural production in China, the majority of rural households have limited access to effective information on fiscal subsidies due to lower levels of education, information technology and digitalisation. As a result, such tax subsidies tend to flow to agricultural enterprises and farmers' cooperatives, among other entities involved in the circulation of agricultural products, thereby reducing the proactive involvement of farmers in production. On the other hand, government support could foster increased dependency among agricultural product circulation enterprises, potentially stifling their impetus and willingness to engage in environmentally conscious production practices. It is therefore recommended that direct price subsidies are reduced while the focus of fiscal support is shifted from the distribution sector to the production sector. This strategic shift can be realised through the formulation of long-term agri-environmental plans aimed at increasing the green total factor productivity in the circulation of agricultural products. Ultimately, government intervention in the agricultural product cycle market should be minimised. Instead of direct intervention, emphasis should be

placed on cultivating a market environment conducive to the circulation of agricultural products and maintaining market order. These measures will encourage farmers to seek more productive production strategies to increase profitability within the context of a market economy [30].

There is a negative correlation between the prices of agricultural products and the total factor productivity of agricultural products, which passes the significance test at the 5% level. This finding also confirms the emergence of problems such as "difficulty in buying and selling agricultural products" and "fluctuating garlic prices" in recent years. In the current development process of China's agricultural product circulation system, some agricultural products cannot fully participate in market competition due to their seasonality and vulnerability, which aggravates the gap between consumers' "shopping baskets" and farmers' "income bags". Ensuring stable agricultural production and prices is therefore a key challenge for improving the efficiency of the agricultural product cycle.

There is a negative correlation between the industrial structure and the green total factor productivity of agricultural products, which passes the significance test at the 5% level. On the one hand, the lagging development of the agricultural industry compared to other industries may imply the relatively backward technologies used in agricultural production. Green technology innovation can improve the efficiency of resource use, but the introduction and promotion of these technologies is influenced by the industrial structure. On the other hand, societal demand for agricultural products and government subsidies may lead to a bias towards the agricultural sector. However, an excessive allocation of resources to agriculture may weaken the competitiveness of other industries and limit the improvement of the green total factor productivity of agricultural products.

The level of foreign direct investment (FDI) is positively correlated with agricultural green total factor productivity in agricultural circulation, and this relationship has been validated at an exact level of 1%. There are several reasons for this positive correlation. Firstly, foreign-invested enterprises often bring advanced technologies and management expertise. By introducing these technologies and experiences, the agricultural circulation sector can benefit from technological advancements and improved management practices, thereby enhancing the level of green total factor productivity. Secondly, foreign investment can promote marketisation and internationalisation in the agricultural circulation sector. Foreign-invested enterprises typically possess extensive market networks and stronger international business experience, which can drive the development of agricultural trade, improve circulation efficiency and enhance the quality of agricultural products. Furthermore, foreign-invested enterprises often have stronger financial capabilities and broader social resources. This enables them to contribute to capital investment and infrastructure development in the agricultural circulation sector, thereby improving the efficiency and quality of agricultural circulation. In summary, the positive correlation between the level of foreign direct investment and agricultural green total factor productivity in agricultural circulation can be attributed to the introduction of advanced technologies, improved management practices, market expansion, international trade facilitation and increased capital investment. These factors collectively contribute to the enhancement of agricultural circulation efficiency and quality.

### 5.2. Spatial Autocorrelation Test

Before conducting the spatial econometric analysis, we first need to examine the existence of spatial correlation between the green total factor productivity of agricultural product circulation and each explanatory variable in the Belt and Road region. If the global Moran's I index is close to 1, it indicates that the green total factor productivity of the agricultural product cycle and each explanatory variable have a positive spatial correlation, which means that high-value areas are often surrounded by other high-value areas. Conversely, when the global Moran's I index is close to −1, it indicates negative spatial correlation, meaning that low-value areas are surrounded by other low-value areas. When the global Moran's I index is close to 0, it indicates a random spatial distribution.

The results, as shown in the Table 6, indicate that the global Moran's I index tests for GTFPCH, environmental regulation, government support, level of foreign direct investment (FDI), industrial structure and prices of agricultural products in the Belt and Road region all have p-values less than 0.1. This suggests that there is a spatial correlation between the green total factor productivity of agricultural product circulation and the relevant explanatory variables.

**Table 6.** Global Moran's I test.

| Variable | Moran's I Coefficient | Expected | Variance | Z-Score | *p*-Value |
|---|---|---|---|---|---|
| GTFPCH | −0.097 | −0.005 | 0.066 | −1.395 | 0.081 * |
| ER | 0.658 | −0.005 | 0.067 | 9.932 | 0.000 *** |
| GP | 0.590 | −0.005 | 0.067 | 8.844 | 0.000 *** |
| FDI | 0.757 | −0.005 | 0.068 | 11.279 | 0.000 *** |
| ST | 0.767 | −0.005 | 0.068 | 11.381 | 0.000 *** |
| AP | 0.374 | −0.005 | 0.068 | 5.598 | 0.000 *** |

Note: *, *** denote significant at 10% and 1% levels.

### 5.3. LM Spatial Econometric Model Test

Before conducting the regression analysis using the spatial Durbin model, this study followed the approach proposed by Lesage [28] et al. and performed Lagrange multiplier (LM) tests, along with the test's robust form (Robust LM), on the residuals to examine the presence of spatial autocorrelation among the green total factor productivity of agricultural product circulation and its explanatory variables under the geographic adjacent spatial weight matrix. Based on the results presented in Table 7, the spatial Durbin model was chosen as the appropriate spatial econometric model for research and analysis, considering the geographic adjacent spatial weight matrix.

**Table 7.** LM spatial econometric model test.

| Statistic | Coefficient | *p*-Value |
|---|---|---|
| LM test no spatial error | 8.3419 *** | 0.0039 |
| Robust LM test no spatial error | 8.1357 *** | 0.0043 |
| LM test no spatial lag | 8.3425 *** | 0.0154 |
| Robust LM test no spatial lag | 8.3425 *** | 0.0154 |

Note: *** denotes significant at 1% levels.

### 5.4. Spatial Durbin Model Regression Analysis

The explained variable, the green total factor productivity of agricultural product circulation, shows a spatial autocorrelation coefficient of 0.0682 with a *p*-value of 0.017 under the geographically adjacent spatial weight matrix, passing the 5% significance level test. This indicates a significant positive spatial spillover effect of the green total factor productivity of agricultural product circulation on itself. It suggests that among the provinces along the Belt and Road, different regions have achieved efficient and coordinated development of agricultural product circulation across regions based on resource integration, which confirms Hypothesis 2: there is a "spatial spillover effect" in the development of agricultural product circulation efficiency among the provinces along the Belt and Road within the framework of green total factor productivity.

The spatial regression coefficients of the explanatory variables, namely, government support and foreign direct investment (FDI) level, also pass the 5% and 1% significance level tests, respectively, under the geographic adjacent spatial weight matrix. The results from the spatial Durbin model are consistent with those from the dynamic panel system GMM estimation. This indicates that both government support and the FDI level can promote the development of high-quality agricultural product circulation in different provinces within the region (Table 8).

**Table 8.** The results of spatial Durbin model regression analysis.

| Variable | Coefficient | *p*-Value |
|---|---|---|
| W × GTFPCH | 0.0682 ** | 0.017 |
| ER | 0.00002 | 0.982 |
| W × ER | 0.00003 | 0.877 |
| GP | −0.00002 | 0.106 |
| W × GP | −0.00002 ** | 0.037 |
| FDI | 0.0001 | 0.180 |
| W × FDI | 0.0001 * | 0.096 |
| ST | 0.1438 | 0.430 |
| W × ST | 0.2387 | 0.796 |
| AP | 0.0035 | 0.273 |
| W × AP | 0.0036 | 0.366 |
| Sigma2_e | 0.0006 *** | 0.000 |
| Log-likelihood | 217.6703 | |
| $R^2$ | 0.1120 | |

Note: *, ** and *** denote significant at 10%, 5% and 1% levels.

## 6. Conclusions and Policy Implications

Inspired by the construction methodology of the green total factor productivity evaluation system, this study integrates the $CO_2$ and COD emissions that were generated in the agricultural product circulation stage as undesired outputs into the evaluation indicator system for agricultural product circulation efficiency along China's Belt and Road Initiative. This integration is complemented by measurements using the SBM model and the GML index. The results show an overall negative growth trend in green total factor productivity of agricultural product circulation in provinces along the Belt and Road, which is particularly evidenced after the launch of the Belt and Road Initiative in 2014. Using the dynamic panel system GMM model and the spatial Durbin model to explore the mechanism of change in the green total factor productivity of agricultural product circulation along the Belt and Road, the following conclusions are drawn: First, there is a significant negative correlation between the level of development of green total factor productivity in the circulation of agricultural products and the lagged period, the same as environmental regulations, government support, agricultural product prices and industrial structure. Second, there is a significant positive correlation between the level of foreign direct investment and the level of green total factor productivity in the cycle of agricultural products. Third, using a spatial weighting matrix based on geographical adjacency, a significant positive spatial spillover effect is identified in the green total factor productivity of the agricultural product cycle. In addition, government support and the level of foreign direct investment can effectively promote the high-quality development of agricultural product circulation in provinces along the Belt and Road. The above analysis results highlight problems within China's agricultural product circulation, including insufficient development of overall green technological innovation, an underdeveloped agricultural product circulation system and limited market competitiveness of agricultural products.

It should be recognised that both green development and sustainable development inherently involve externalities, implying that the current development in a certain region may come at the cost of sacrificing its own future development, at the expense of the current or future development of other regions or even at the simultaneous cost of sacrificing the future development of both local and other regions. This study takes the external costs as internalised, and the results just illustrate the inherent problem as reflected by the presented negative growth and negative correlation. Therefore, within the context of the Belt and Road Initiative, a nuanced perspective is required when addressing the issues of green and sustainable development or rapid development in agricultural product circulation. The policy insights for decision-makers are as follows.

### 6.1. Enhance Technological Innovation and Improve Green Total Factor Productivity

Firstly, optimise resource allocation and improve the efficiency of resource allocation in the circulation of agricultural products. This can be achieved by optimising land use, water resource management, energy use and other aspects to increase productivity and resource use efficiency.

Secondly, promote the modernisation of agriculture by increasing technological investment and promoting agricultural modernisation and digital transformation. This can occur within the framework of promoting a dual-circulation development pattern that integrates domestic and overseas markets and leverages technologies such as the Internet of Things, big data and artificial intelligence to integrate "Internet+" with agricultural product circulation. This approach can reduce the cost of agricultural product circulation and expand distribution channels.

Thirdly, promote sustainable agricultural practices by advocating sustainable agricultural practices, including organic farming, restoration and conservation of agricultural ecosystems and climate-smart agriculture. Through environmentally friendly agricultural production methods, the environmental impact is reduced and production efficiency is increased.

### 6.2. Optimise Environmental Regulations and Establish Interdepartmental Coordination Mechanisms

Firstly, the Chinese government and relevant departments should develop scientific and rational environmental regulatory measures that balance the relationship between environmental protection and the development of the agricultural product circulation industry. It is important to ensure that regulatory measures meet the requirements of green production and circulation, encourage enterprises to adopt sustainable production methods and provide necessary technological and financial support.

Secondly, establishing interdepartmental coordination mechanisms for the development of agricultural product circulation is crucial to strengthen policy alignment and resource integration. This includes increasing financial support for agricultural product circulation enterprises, such as loans, venture capital and innovative financial products. Encouraging financial institutions to innovate financial services by providing financing facilities and risk management tools can effectively promote the improvement of green total factor productivity, the utilisation of spatial spillover effects and the development of the green agricultural product circulation industry.

Thirdly, the government should strengthen guidance and supervision of the agricultural product circulation industry to promote standardised development. This can be achieved by enhancing market access management, product quality monitoring and information disclosure while safeguarding market order and consumer rights.

### 6.3. Ensure Stable Agricultural Production and Prices

Continue to promote structural reform on the supply side of agricultural products, firstly by emphasising the key role of circulation units represented by farmers. Mobilise their production enthusiasm and enhance rural informationisation by introducing specialised talent and establishing information exchange platforms. This approach will prevent indiscriminate production and promote the standardised and scaled development of agricultural production.

Secondly, make full use of the market's ability to allocate resources. Starting from the demand side for agricultural products, adjust production structures and methods and avoid redundant competition in order to safeguard agricultural product prices.

Thirdly, speed up the development of a modern agricultural transport and logistics system. Emphasis should be placed on the development and use of cold chain logistics in the circulation of agricultural products. These efforts aim to reduce the rate of spoilage during transport and ensure consistent product quality throughout the transport process.

### 6.4. Establish Green Demonstration Zones and Promote Coordinated Development between Regions

As the world economy develops rapidly and globalisation accelerates, the establishment of demonstration zones can bring several benefits. First, it enables the complementary use and integration of resources, attracts investment and talent and generates a spillover effect to stimulate the development of surrounding areas. Second, it can strengthen economic

cooperation and coordination among countries along the route, facilitating the sharing of management experience and technological innovation to maximise spatial spillovers. Third, in promoting the Belt and Road Initiative, it is crucial to attach great importance to the diverse cultures and historical backgrounds of countries along the route, and to establish cooperative relations based on mutual respect, equality and mutual benefit. By aligning with the United Nations' 2030 Sustainable Development Agenda and promoting green and low-carbon transformation, new vitality can be injected into the joint development of a community with a shared future for people and nature.

**Author Contributions:** Conceptualization, M.D. and G.W.; methodology, G.W.; software, J.W. and G.W.; investigation, Q.L. and Y.G.; writing—original draft preparation, M.D., G.W. and Y.G. All authors have read and agreed to the published version of the manuscript.

**Funding:** Project of Liaoning Provincial Department of Education: Research on collaborative innovation of the whole industrial chain of light industry based on quality development J2020084; Project of Liaoning Provincial Federation of Social Sciences: Proposals for developing high-tech service industry and enhancing its role in supporting economic development 2021lslzdwtkt-01; National Nature Fund Project: Research on technological innovation and productivity improvement in the real economy from the perspective of financialisation 71703012; Liaoning Provincial Education Department Scientific Research Project: Economic uncertainty, financialisation of the real economy and capital allocation efficiency J202106; Dalian Social Science Federation Project: Developing modern agricultural product circulation system and promoting high-quality agricultural development in Dalian 2023dlskzd012.

**Institutional Review Board Statement:** Not applicable.

**Informed Consent Statement:** Not applicable.

**Data Availability Statement:** Not applicable.

**Conflicts of Interest:** The authors declare no conflict of interest.

## Appendix A

GML index decomposition formula:

$$GML^{t,t+1}\left(x^{t+i},y^{t+i},z^{t+i};x^t,y^t,z^t\right) = \frac{1+D^G\left(x^t,y^t,z^t\right)}{1+D^G\left(x^{t+i},y^{t+i},z^{t+i}\right)}$$

$$= \frac{1+D^t\left(x^t,y^t,z^t\right)}{1+D^{t+i}\left(x^{t+i},y^{t+i},z^{t+i}\right)} \times \frac{\frac{1+D^G\left(x^t,y^t,z^t\right)}{1+D^t\left(x^t,y^t,z^t\right)}}{\frac{1+D^G\left(x^t,y^t,z^t\right)}{1+D^{t+i}\left(x^{t+i},y^{t+i},z^{t+i}\right)}}$$

$$= GEC^{t+i} \times GTC^{t+i}$$

## Appendix B

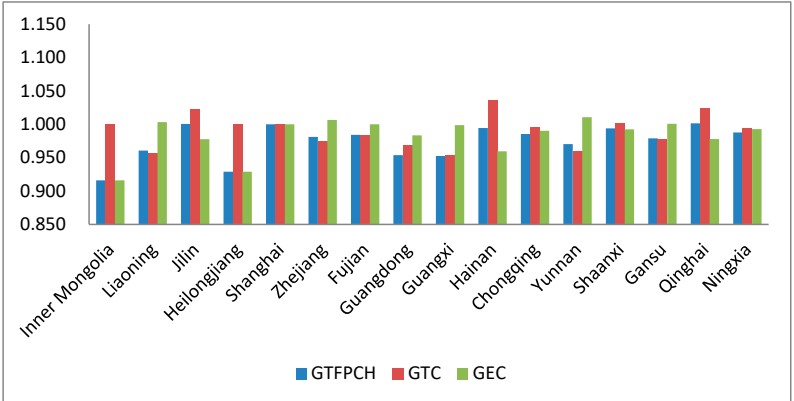

**Figure A1.** Changes in relevant indices of provinces along the Belt and Road Initiative before its introduction, 2014–2021. Source of data: actual measurements.

**Table A1.** Share of China's agricultural output in the value of total output, 2010–2021 (unit: %).

| | 2010 | 2011 | 2012 | 2013 | 2014 | 2015 | 2016 | 2017 | 2018 | 2019 | 2020 | 2021 |
|---|---|---|---|---|---|---|---|---|---|---|---|---|
| **Beijing** | 0.8811 | 0.8385 | 0.8401 | 0.8299 | 0.7454 | 0.6092 | 0.5056 | 0.4298 | 0.3915 | 0.3214 | 0.2981 | 0.2764 |
| **Tianjin** | 1.5782 | 1.4125 | 1.3309 | 1.3114 | 1.2711 | 1.2627 | 1.2313 | 0.9109 | 0.9182 | 1.3133 | 1.4924 | 1.4361 |
| **Hebei** | 12.5663 | 11.8525 | 11.9912 | 12.3684 | 11.7176 | 11.5394 | 10.8911 | 9.2014 | 9.2696 | 10.0228 | 10.7166 | 9.9781 |
| **Shanxi** | 6.0264 | 5.7078 | 5.7651 | 6.1403 | 6.1818 | 6.1345 | 6.0135 | 4.6313 | 4.4038 | 4.8437 | 5.3630 | 5.6967 |
| **Inner Mongolia** | 9.3838 | 9.0969 | 9.1217 | 9.5020 | 9.1606 | 9.0706 | 9.0323 | 10.2494 | 10.1440 | 10.8246 | 11.6656 | 10.8471 |
| **Liao Ning** | 8.8371 | 8.6183 | 8.6766 | 8.5740 | 7.9847 | 8.3157 | 9.7679 | 8.1262 | 8.0319 | 8.7427 | 9.0966 | 8.9247 |
| **Jilin** | 12.1158 | 12.0869 | 11.8275 | 11.6269 | 11.0410 | 11.3508 | 10.1410 | 7.3295 | 7.7000 | 10.9776 | 12.6144 | 11.7396 |
| **Heilongjiang** | 12.5658 | 13.5233 | 15.4377 | 17.4985 | 17.3635 | 17.4593 | 17.3563 | 18.6462 | 18.3415 | 23.3786 | 25.0998 | 23.2741 |
| **Shanghai** | 0.6650 | 0.6509 | 0.6332 | 0.5985 | 0.5272 | 0.4371 | 0.3885 | 0.3616 | 0.3194 | 0.2723 | 0.2676 | 0.2314 |
| **Jiang Su** | 6.1317 | 6.2406 | 6.3233 | 6.1629 | 5.5837 | 5.6849 | 5.2685 | 4.7108 | 4.4729 | 4.3122 | 4.4166 | 4.0583 |
| **Zhejiang** | 4.9078 | 4.8982 | 4.8114 | 4.7503 | 4.4238 | 4.2739 | 4.1590 | 3.7357 | 3.5002 | 3.3638 | 3.3572 | 3.0049 |
| **Anhui** | 13.9896 | 13.1714 | 12.6582 | 12.3331 | 11.4750 | 11.1639 | 10.5202 | 9.5576 | 8.7914 | 7.8561 | 8.2333 | 7.8228 |
| **Fujian** | 9.2533 | 9.1812 | 9.0180 | 8.8986 | 8.3755 | 8.1529 | 8.2026 | 6.8831 | 6.6468 | 6.1239 | 6.2234 | 5.9366 |
| **Jiangxi** | 12.7706 | 11.8866 | 11.7402 | 11.4133 | 10.7143 | 10.6016 | 10.2953 | 9.1734 | 8.5392 | 8.3109 | 8.7250 | 7.8809 |
| **Shandong** | 9.1608 | 8.7603 | 8.5611 | 8.6727 | 8.0744 | 7.9030 | 7.2461 | 6.6535 | 6.4738 | 7.1994 | 7.3347 | 7.2555 |
| **Henan** | 14.1090 | 13.0416 | 12.7352 | 12.6228 | 11.9068 | 11.3765 | 10.5906 | 9.2907 | 8.9258 | 8.5431 | 9.7346 | 9.5450 |
| **Hubei** | 13.4460 | 13.0871 | 12.8032 | 12.5592 | 11.6033 | 11.2007 | 11.2025 | 9.9469 | 9.0115 | 8.3117 | 9.5110 | 9.3210 |
| **Hunan** | 14.5000 | 14.0727 | 13.5604 | 12.6491 | 11.6459 | 11.5272 | 11.3414 | 8.8441 | 8.4654 | 9.1742 | 10.1491 | 9.3847 |
| **Guang Dong** | 4.9703 | 5.0088 | 4.9892 | 4.9024 | 4.6701 | 4.5947 | 4.5691 | 4.0259 | 3.9387 | 4.0413 | 4.3066 | 4.0232 |
| **Guangxi** | 17.5035 | 17.4665 | 16.6655 | 16.2997 | 15.3988 | 15.2677 | 15.2683 | 15.5388 | 14.8354 | 15.9520 | 16.0485 | 16.2302 |
| **Hainan** | 26.1482 | 26.1323 | 24.9179 | 24.0419 | 23.1244 | 23.0833 | 23.3976 | 21.5761 | 20.6974 | 20.3499 | 20.5333 | 19.3724 |
| **Chongqing** | 8.6477 | 8.4356 | 8.2388 | 8.0332 | 7.4392 | 7.3177 | 7.3461 | 6.5694 | 6.7685 | 6.5722 | 7.2125 | 6.8904 |
| **Sichuan** | 14.4476 | 14.1892 | 13.8116 | 13.0446 | 12.3737 | 12.2360 | 11.9307 | 11.5260 | 10.8822 | 10.3125 | 11.4336 | 10.5140 |
| **Guizhou** | 13.5812 | 12.7366 | 13.0164 | 12.8522 | 13.8182 | 15.6210 | 15.6766 | 15.0085 | 14.5851 | 13.5996 | 14.2477 | 13.9428 |
| **Yunnan** | 15.3426 | 15.8663 | 16.0488 | 16.1706 | 15.5297 | 15.0948 | 14.8434 | 14.2790 | 13.9749 | 13.0798 | 14.6763 | 14.2566 |
| **Shanxi** | 9.7639 | 9.7576 | 9.4797 | 9.5109 | 8.8465 | 8.8650 | 8.7314 | 7.9523 | 7.4890 | 7.7188 | 8.6607 | 8.0850 |
| **Gansu** | 14.5430 | 13.5199 | 13.8137 | 14.0295 | 13.1751 | 14.0507 | 13.6575 | 11.5250 | 11.1727 | 12.0491 | 13.2880 | 13.3229 |
| **Qinghai** | 9.9909 | 9.2838 | 9.3428 | 9.8803 | 9.3747 | 8.6440 | 8.5983 | 9.0829 | 9.3570 | 10.1789 | 11.1214 | 10.5391 |
| **Ningxia** | 9.4274 | 8.7594 | 8.5167 | 8.6930 | 7.8845 | 8.1655 | 7.6248 | 7.2779 | 7.5529 | 7.4678 | 8.6215 | 8.0601 |
| **Xinjiang** | 19.8370 | 17.2318 | 17.5951 | 17.5628 | 16.5914 | 16.7197 | 17.0883 | 14.2607 | 13.8706 | 13.1039 | 14.3596 | 14.7407 |

Source of data: "China Statistical Yearbook (2021–2022)".

**Table A2.** Percentage of China's population employed in agriculture, 2010–2021 (unit: %).

| | 2010 | 2011 | 2012 | 2013 | 2014 | 2015 | 2016 | 2017 | 2018 | 2019 | 2020 | 2021 |
|---|---|---|---|---|---|---|---|---|---|---|---|---|
| **Beijing** | 14.400 | 13.800 | 13.800 | 13.700 | 13.650 | 13.500 | 13.500 | 13.500 | 13.500 | 13.400 | 12.960 | 12.520 |
| **Tianjin** | 20.745 | 19.500 | 18.450 | 17.990 | 17.730 | 17.360 | 17.070 | 17.070 | 16.850 | 16.520 | 15.835 | 15.150 |
| **Hebei** | 55.700 | 54.400 | 53.200 | 51.880 | 50.670 | 48.670 | 46.680 | 44.990 | 43.570 | 42.380 | 40.620 | 38.860 |
| **Shanxi** | 52.165 | 50.320 | 48.740 | 47.440 | 46.210 | 44.970 | 43.790 | 42.660 | 41.590 | 40.450 | 38.515 | 36.580 |
| **Inner Mongolia** | 44.990 | 43.380 | 42.260 | 41.290 | 40.490 | 39.700 | 38.810 | 37.980 | 37.290 | 36.630 | 34.210 | 31.790 |
| **Liao Ning** | 37.800 | 35.950 | 34.350 | 33.550 | 32.950 | 32.650 | 32.630 | 32.510 | 31.900 | 31.890 | 29.540 | 27.190 |
| **Jilin** | 46.640 | 46.600 | 46.300 | 45.800 | 45.190 | 44.690 | 44.030 | 43.350 | 42.470 | 41.730 | 39.180 | 36.630 |
| **Heilongjiang** | 44.000 | 43.500 | 43.100 | 42.600 | 41.990 | 41.200 | 40.800 | 40.600 | 39.900 | 39.100 | 36.700 | 34.300 |
| **Shanghai** | 11.050 | 10.700 | 10.700 | 10.400 | 10.400 | 12.400 | 12.100 | 12.300 | 11.900 | 11.700 | 11.195 | 10.690 |
| **Jiang Su** | 41.250 | 38.100 | 37.000 | 35.890 | 34.790 | 33.480 | 32.280 | 31.240 | 30.390 | 29.390 | 27.725 | 26.060 |
| **Zhejiang** | 39.900 | 37.700 | 36.800 | 36.000 | 35.130 | 34.200 | 33.000 | 32.000 | 31.100 | 30.000 | 28.670 | 27.340 |
| **Anhui** | 56.550 | 55.200 | 53.500 | 52.140 | 50.850 | 49.500 | 48.010 | 46.510 | 45.310 | 44.190 | 42.395 | 40.600 |
| **Fujian** | 45.250 | 41.900 | 40.400 | 39.230 | 38.200 | 37.400 | 36.400 | 35.200 | 34.180 | 33.500 | 31.905 | 30.310 |
| **Jiangxi** | 55.560 | 54.300 | 52.490 | 51.130 | 49.780 | 48.380 | 46.900 | 45.400 | 43.980 | 42.580 | 40.560 | 38.540 |
| **Shandong** | 50.365 | 49.050 | 47.570 | 46.250 | 44.990 | 42.990 | 40.980 | 39.420 | 38.820 | 38.490 | 37.275 | 36.060 |
| **Henan** | 60.865 | 59.430 | 57.570 | 56.200 | 54.800 | 53.150 | 51.500 | 49.840 | 48.290 | 46.790 | 45.170 | 43.550 |
| **Hubei** | 51.085 | 48.170 | 46.500 | 45.490 | 44.330 | 43.150 | 41.900 | 40.700 | 39.700 | 39.000 | 37.460 | 35.920 |
| **Hunan** | 55.850 | 54.900 | 53.350 | 52.040 | 50.720 | 49.110 | 47.250 | 45.380 | 43.980 | 42.780 | 41.535 | 40.290 |
| **Guang Dong** | 35.050 | 33.500 | 32.600 | 32.240 | 32.000 | 31.290 | 30.800 | 30.150 | 29.300 | 28.600 | 26.985 | 25.370 |
| **Guangxi** | 59.500 | 58.200 | 56.470 | 55.190 | 53.990 | 52.940 | 51.920 | 50.790 | 49.780 | 48.910 | 46.920 | 44.930 |
| **Hainan** | 50.185 | 49.500 | 48.400 | 47.260 | 46.240 | 44.880 | 43.220 | 41.960 | 40.940 | 40.770 | 39.895 | 39.020 |
| **Chongqing** | 46.695 | 44.980 | 43.020 | 41.660 | 40.400 | 39.060 | 37.400 | 35.920 | 34.500 | 33.200 | 31.435 | 29.670 |
| **Sichuan** | 59.735 | 58.170 | 56.470 | 55.100 | 53.700 | 52.310 | 50.790 | 49.210 | 47.710 | 46.210 | 44.195 | 42.180 |
| **Guizhou** | 67.575 | 65.040 | 63.590 | 62.170 | 59.990 | 57.990 | 55.850 | 53.980 | 52.480 | 50.980 | 48.320 | 45.660 |
| **Yunnan** | 64.600 | 63.200 | 60.690 | 59.520 | 58.270 | 56.670 | 54.970 | 53.310 | 52.190 | 51.090 | 50.025 | 48.960 |
| **Shanxi** | 54.600 | 52.700 | 49.979 | 48.689 | 47.429 | 46.080 | 44.660 | 43.210 | 41.870 | 40.570 | 38.470 | 36.370 |
| **Gansu** | 65.100 | 62.850 | 61.250 | 59.870 | 58.320 | 56.810 | 55.310 | 53.610 | 52.310 | 51.510 | 49.090 | 46.670 |
| **Qinghai** | 55.940 | 53.780 | 52.560 | 51.490 | 50.220 | 49.700 | 48.370 | 46.930 | 45.530 | 44.480 | 41.770 | 39.060 |
| **Ningxia** | 52.040 | 50.180 | 49.330 | 47.990 | 46.390 | 44.770 | 43.710 | 42.020 | 41.120 | 40.140 | 37.035 | 33.930 |
| **Xinjiang** | 58.305 | 56.460 | 56.020 | 55.530 | 53.930 | 52.770 | 51.650 | 50.620 | 49.090 | 48.130 | 45.445 | 42.760 |

Source of data: "China Statistical Yearbook (2021–2022)".

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
