# Peer review of "Study of the Spatial Spillover Effects of the Efficiency of Agricultural Product Circulation in Provinces along the Belt and Road under the Green Total Factor Productivity Framework"

_sustainability, doi:10.3390/su151612560_

Round 1

Reviewer 1 Report

The writing of this paper requires significant revisions. For instance, the author introduces some content in the methodology section that seems more appropriate for the literature review section (3.2 and 3.3). In the results section, the author also includes information that should have been clarified in the methodology section, such as calculation formulas and data sources. Please refer to the writing style of authoritative journal papers and move the content of each section to the chapters where they are more appropriate.

By the way, if the calculation methods were not proposed by you, please cite the source after the formula. 

Additionally, I suggest merging the content of Section 4 and 5 into a single section called "Results" to present them as a cohesive whole.

In terms of data processing, please upload the secondary data that you extracted from various databases as attachments. Furthermore, the inclusion of a significance level of 0.1 does not adhere to general standards. Path relationships greater than 0.05 should be considered non-significant. 

Lastly, I recommend splitting the "Suggestions" section into two chapters: "Discussion" and "Conclusion." In the discussion section, you should present your recommendations based on the results of each model or path relationship, including theoretical and practical suggestions. Please approach the writing in a critical manner rather than a descriptive one. In the conclusion section, you need to clarify your research contributions, limitations, and future research directions.

Author Response

Dear reviewer:
Thank you for taking the time to review this manuscript. I really appreciate all your comments and suggestions! For my revision notes, I would be grateful if you could consult the Word document I have submitted.

Reviewer 2 Report

Please read the comment.

Please read the comment.

Author Response

(The authors gave the same response as above.)

Reviewer 3 Report

This paper studies agricultural productivity in provinces along the Belt and Road. In overall, the topic has a good relevance and could be of interest for the journal, but the paper has serious methodological shortcomings that make it unsuitable for publication. My concerns are detailed below.

It is not completely clear what green TFP means. The authors talks about it as a well established framework, but it is not so. Therefore, the concept of green TFP and, more important, its measurement, must be explained and referenced. Why DEA with directional distance functions (DDFs) is a proper method to measure green TFP change? The explanation at lines 213-220 is not sufficient. Is there any other method? Please, explain. Also the abstract requires a brief explanation.

Often, even in the title of the manuscript, the term 'efficiency' is used as a synonym of productivity, but this is not correct. As formalized by the Malmquist indices, efficiency change is just a component of productivity change, and they are equal only in case of constant technological level.

The methods used are not sufficiently referenced and presented. In Section 3.1, it is necessary to add formal details of DEA with DDFs, as well as of GML indices and their decomposition into GTC and GEC components.

Section 4: The use of DEA scores in subsequent regression analysis is well known to entail bias. Several methods have been proposed to avoid this, see, e.g., Simar & Wilson (2007):

Simar L, Wilson PW (2007). “Estimation and Inference in Two-Stage, Semi-Parametric Models of
Production Processes.” Journal of Econometrics, 136(1): 31-64.

Dynamic panel model in Section 4:
- The acronyms of the variables in the model must be explained before the formula.
- In the formula, I think that parameter alpha_0 should have subscript i (one intercept for each unit).
- Add the assumptions on the error terms epsilon.
- Why did you include the second lag of the response without the first? It is pretty strange.
- Why did you assume only contemporaneous effects of the explanatory variables?
- The model includes autoregressive terms, therefore autocorrelated errors would make the estimation inconsistent: it is important to check the ACF of residual.

Table 4.2: for better readability, it is sufficient to use 4 decimals.

Section 5: I do not agree with the approach to run the dynamic panel model without spatial effects and then to check for spatial autocorrelation. If spatial correlation is supposed, then it should be checked before running the regression model, and, if confirmed, spatial lags must be included in the model, e.g., by using the spatial Durbin model.

Author Response

(The authors gave the same response as above.)

Reviewer 4 Report

Dear Author,

the study deals with the interesting topic of the impact of infrastructure development on the growth and development of production and economic integration between regions. Nevertheless, many inconsistencies in this manuscript must be clarified/resolved for it to be of scientific value.

1. The Belt and Road infrastructure is mainly used to enable international cooperation and integration; it is not clear why this should affect the development of agricultural trade, especially when the size of agricultural production, surpluses, infrastructure for processing and preparation of products for marketing and long-term transport are not considered.
2. It is not explained why the increase in logistical and transport activity is thought to lead to environmental reductions. Currently, there is a desire to develop short food chains to reduce the unnecessary two-way circulation of goods.
3. It is not justified in the work why foreign trade in food, in general, is considered, including from provinces other than Belt and Road, and it is supposed to have an impact on the provinces included in the study. Are these provinces with a significant share of food products for export? Or whether they are provinces with significant food shortages.
4.
Are the examined regions important in the national system of agricultural products circulation. If not, how the infrastructure created for other purposes should, according to the authors, affect the emissions reduction. Suppose the infrastructure is mainly used for other purposes. In that case, its development, with a constant volume of agricultural production, will lead to the deterioration of efficiency indicators based on the volume of agricultural production.
5. The above should be considered in terms of the aim of the study.
6. The literature review largely concerns such issues as: product losses, quality standards, scale of production, scale of processing, infrastructure, sustainable development of agriculture. Are such aspects taken into account in the study?
7.
The hypotheses put forward are not fully verified in the work, has it been established that there is sustainable development of agriculture, and effective circulation of agricultural products, have there been changes in the spatial connection of the province before and after the investment? It is unclear whether the introduction of pro-environmental regulations is considered a factor directly affecting the economy without considering the real costs of regulation burdening enterprises and increasing costs.
8.
due to the lack of direct data, the data used are calculated indirectly. This means that, for example, agribusiness is burdened with infrastructure costs proportional to its share in food consumption or exports. Where and how the GHG emissions were determined.
9. Regarding the results, it seems that the suggestions presented are of a general nature. It is reported that better use of land, water, energy, and inputs can be made to achieve better results. It is proposed to introduce IoT, AI, Big Data and even organic farming solutions in agriculture and agribusiness. These results bear little relevance to the target and hypotheses, and are too vague.

I am convinced that the presented work should not be published in its current form. It requires clarification in terms of what is being studied and the selection of variables. Maybe first check if there was a spatial correlation and how it changed using the Moran Index for different periods.

The English language at work is understandable and without significant errors. There are single typos and single sentences with an unclear message. Nevertheless, in the context of other sentences, the text is understandable to the reader, and the message is clear.

Author Response

(The authors gave the same response as above.)

Round 2

Reviewer 1 Report

I am very disappointed with the revised version.    The writing issues I pointed out in the previous round have not been addressed.    For instance, this paper includes two stages of research, but in the Research Method section, the author still only presents the content of Stage 1.    The research content and results are mixed together in the section the author claims to be the methodology.    Additionally, the paper lacks a Discussion section.

The author has chosen to ignore my suggestion.    Given the author's casual attitude, I am deeply concerned about the academic rigor and thoroughness of this paper.    I am sorry, but I cannot recommend the publication of this manuscript.

Author Response

Dear reviewer:
Thank you for taking the time to review this manuscript. I really appreciate all your comments and suggestions! For my revision notes, I would be grateful if you could consult the PDF document I have submitted.

Reviewer 2 Report

Again, long texts create confusion and boredom around the paper topic: the paper mentioned two hypotheses. I do not know whether this type of journal's authors and readers prefer long, repeated texts. In my view, the paper texts should be shortened, taking only the relevant texts. 

Delete unnecessary words or phrases as many as possible.

Some parts of English texts need attention in academic writing.

Author Response

(The authors gave the same response as above.)

Reviewer 3 Report

The manuscript has reached a sufficient level of quality after the revisions. I have just few minor issues to suggest.

Abstract: the acronyms 'SBM model', 'GML index' and 'system GMM' must be defined. In particular, 'SBM' is still undefined at line 247 (the first definition is at line 263).

Eq. 2: Also the components GTFPCH, GTC and GEC that appear in the results (Table 1) must be defined here. Make sure to uniform the notation.

Section 4.1 has no introduction and starts directly with a subsection. It would increase clarity if the section would begin with the content of the various subsections (4.1.1, 4.1.2, 4.1.3). The same for Section 4.2.

'System GMM'  appears for the first time at line 443 but it is defined only at line 508. Important: I did not find any motivation for the use of this method. Why did you use GMM rather than classic OLS? This should be added along with Eq. 3.

Line 522: pay attention to the fact that system GMM is not a model, but an estimation method for the regression model in Eq. 3.

Author Response

(The authors gave the same response as above.)

Reviewer 4 Report

Dear Authors,
the revised version of the manuscript presents the issue under investigation more clearly. However, there is still no strong case for how BRI affects agricultural production circulation and what conditions are needed for food to flow more efficiently in China (e.g. fewer two-way streams, concentrated storage, etc.). Some explanations are included in the summary. As for the conclusions, it seems that too much attention has been paid to modern telecommunications solutions and e.g. IoT, and little space has been devoted to what technical conditions should be met to make the food flow more environmentally effective.
In my opinion, after some minor changes, the work will be of publishable quality.
It should be emphasized that more attention was paid to the idea of GTFP and measurement methods and less to selecting variables that would adequately explain the observed changes. Certain limitations to the inference resulting from data availability and variables selection should be included in the short Limitations section at the end of the paper.

Some detailed comments are below:
1. It is worth specifying the percentage of agricultural production and the percentage of the Chinese population in the surveyed provinces.
2. In line 134, there is a "relevant suggestion" according to Cheng [6]. What are the proposed measurement methods? Were they used in this work?
3. Some sources or research results should support the EKC hypothesis. Also, in Sustainability, many studies present the EKC observation on the example of economies and sectors in different countries.
4. In section 2.2. no sources are presented (even for M. Porter's theorems.
5. For Equation 2, add explanations for all variables and labels. It can also refer to the variables described in lines 273-279. It is also worth adding a description of the decomposition methods for GTC and GEC (the results are in Table 1).
6. In line 311 is "replaces". Should it be "represents"? Text 310-314 is not clear enough in the variable description.
7. It is worth presenting a bulleted list of variables for GTFP instead of presenting it in descriptive form (lines 309-329).
8. A valuable addition would be to present the characteristics of selected variables in the table (maybe in the annexe). Nothing is known about the studied quantities (see also point 1).
9. In Figure 1, add horizontal grid lines or data for years.
10. Figure 2 is the same as Table 1. Is such duplication of results necessary and acceptable?
11. What does "frontier provinces" mean in the title of Figure 2. Earlier in the text, the provinces located on the Belt and Road are mentioned.
12. In Table 3, enter the units of measurement.
13. Can the average for FDI be lower than the minimum (Table 3).
14. For reasons for the negative impact of subsidies on TFP, see Biagini's work - https://doi.org/10.1016/j.foodpol.2023.102473. She is very inspiring. (lines 557-564, 574-583)
15. The impact of government subsidies should be better explained. One is that they have a limiting effect on the initiative (something like rent-seeking), and the summary is that they have a positive impact (659-661). It should probably be noted that subsidies must target specific activities.
16. The suggestions section is too much about IoT, AI, and VC. It is not explained what this is for. Whether for flow coordination, distribution network planning or whatever.
17. Section 5 is worth adding something about the development of the "cold supply chain" and the preservation of food in processing, which will make food more suitable for long-term storage and transport.
18. I suggest adding a Limitations section where the authors indicate the limitations of their work and limitations connected to the direct application of their proposals.
19. Find and remove typos from the text.

Author Response

(The authors gave the same response as above.)

Round 3

Reviewer 1 Report

This article did not meet my expectations. However, I would not discourage the authors, but would also encourage them to create a broader picture, that is, a more detailed review of the literature, to present contemporary sources of literature, to refer to them and to determine in more detail the research area they are working on. Please continue to strive in the future and make even greater contributions to the academic community. Congratulations.